# Secondary Scoliosis as a Complication of Acute Transverse Myelitis in a Child

**DOI:** 10.3390/jfmk5020039

**Published:** 2020-06-09

**Authors:** Silvia D’Amico, Piero Pavone, Gianluca Testa, Filippo Greco, Lidia Marino, Pierluigi Smilari, Vito Pavone

**Affiliations:** 1Department of Clinical and Experimental Medicine, Section of Pediatrics and Child Neuropsychiatry, University of Catania, Via Santa Sofia 78, 95123 Catania, Italy; silviam.damico@gmail.com (S.D.); ppavone@unict.it (P.P.); coicoico@hotmail.com (F.G.); lidia.m@hotmail.it (L.M.); pierluigismilari@hotmail.it (P.S.); 2Department of Orthopaedics and Traumatologic Surgery, AOU Policlinico-Vittorio Emanuele, University of Catania, Via Santa Sofia, 78, 95123 Catania, Italy; gianpavel@hotmail.com

**Keywords:** Transverse Myelitis, scoliosis, spine, autonomic dysfunction, inflammatory demyelination

## Abstract

Acute transverse myelitis (ATM) is a rare neurological condition that affects the spinal cord. Several events, including infections, autoimmune conditions, inflammatory, and drug-induced factors, may cause this disorder. Correct and rapid etiological diagnosis is necessary in order to start appropriate treatment that mainly consists of immunomodulating therapy, high dose intravenous corticosteroids, and in plasma exchange in noninfectious cases. The outcome is varied and depends on several factors. In children, the prognosis is usually good. We report a case of an 11-year-old boy who presented with interscapular pain, right leg steppage, homolateral hyposthenia of the upper limb, and signs of autonomic dysfunction. After performing specific and instrumental exams, a diagnosis of transverse myelitis was reached, and appropriate therapy was performed. A few days post-treatment, the child developed a secondary scoliosis, involving a thoracolumbar curve with loss of cervical and lumbar lordosis. After rehabilitative treatment was undertaken for 12 months, a complete recovery and normal restoration of spinal physiological curves was obtained. The pediatric cases of ATM have a good response to steroid therapy combined with physiotherapy. Collaboration among the various specialists is worthwhile, in order to lead to a correct and rapid diagnosis.

## 1. Introduction

Acute transverse myelitis (ATM) is a rare neurological disorder that is mainly associated to inflammatory demyelinating and immune-mediated factors and characterized by relatively acute onset of motor, sensory, and autonomic dysfunction. Affected children comprise 20% of the total number of ATM cases [1,2]. The incidence of pediatric ATM is 1.7–2 cases per million children yearly [3]. The male to female ratio is 1.1–1.6:1 [2]. The age distribution of pediatric ATM is bimodal and primarily affects children under 5 and older than 10 years of age. ATM could be caused by infective agents or inflammatory events, such as systemic autoimmune disorders, acquired demyelinating diseases, and/or paraneoplastic syndromes. The disorder may be related to relapsing acquired demyelinating syndromes (ADS), including multiple sclerosis (MS), acute disseminated encephalomyelitis (ADEM), and neuromyelitis optica (NMO). Direct traumatic causes and drug/toxin-induced events have been also reported. Minor trauma for an average of 8 days prior to the onset of neurological symptoms was reported as a cause, and previous infections (66%) or vaccinations (28%) were reported in more of children with ATM. Sensory symptoms consist of burning paresthesia, hyperesthesia, allodynia, and/or numbness. Most children develop urinary retention and need catheterization [2]. ATM patients can present with back pain as the first symptom followed by motor and sensory deficits and/or bladder/bowel dysfunction. Differential diagnosis is mandatory, especially to rule out tumor or vascular disorders, including several parameters: (1) other forms of myelopathy, such as compressive or non-inflammatory disorders, including epidural hematoma, intervertebral disk herniation, vertebral body fracture, ischemic myelopathy due to arterial compromise or venous hypertension, and spinal cord tumors [4]; (2) secondary ATM, including infectious myelitis, rheumatological diseases (such as Systemic Lupus Erythematosus ([SLE) and Sjogren’s syndrome [SS]), paraneoplastic syndromes, demyelinating CNS disease (ADEM, MS, and NMO) [5,6]; and (3) non-myelopathic disorders, such as Guillain-Barre syndrome.

We report a case of an 11-year old boy with ATM and an unusual onset of secondary thoracolumbar scoliosis.

## 2. Case Report

An 11-year old boy was referred to our department in October 2019 because of an incident of domestic trauma in which he fell backwards, which caused interscapular pain with cervical irradiation. Informed consent was obtained by parents. A spine X-ray showed mild right-convex scoliotic attitude, inversion of physiological lordosis, and a hint of anterolisthesis on C3. Blood tests were not abnormal. After an orthopedic consultation, a cervical collar for seven days and analgesic therapy were suggested. He also presented steppage of the right leg and hyposthenia of the upper limb on the same side. Brain and spinal magnetic resonance imaging (MRI) showed inflammation and swelling of the spinal cord, which extended from C2 to C7. He was immediately hospitalized in poor systemic condition.

Upon physical examination, he presented hyperhidrosis, flushing of the face, itching, and a kyphotic position. Upon neurological examination, hyposthenia of the upper limbs bilaterally and uncertain ambulation with steppage of the right lower limb have been noted; meningeal signs were absent, patellar reflexes were hyperelicitable, and sensitivity of lower limbs and other body areas was preserved. Blood and infectious disease tests were repeated, showing results within the normal range. A new spinal MRI highlighted a significant increase in spinal cord size with edematous appearance, characterized by marked hyperintensity in long TR sequences with associated hydrosyringomyelia of discrete entity between C1 and Th3. A lumbar puncture revealed a slight increase in both the number of polymorphonucleated cells and Link index. Cerebrospinal fluid (CSF) for autoimmunity (antibodies anti-MOG, anti-aquaporin, anti GlyR3, and other neuronal antibodies) and oligoclonal bands were negative. The enhanced MRI showed uneven uptake of gadolinium at the lesion level. These data, in association with a laboratory test and lumbar puncture results, indicate inflammatory lesions, and suggest a diagnosis of ATM (Table 1). The patient underwent steroid therapy (intravenous (iv) methylprednisolone boluses 900 mg/day for five days followed by prednisone 15 mg 2 times/day orally, which was then gradually reduced), antibiotic, and antiviral prophylaxis.

Clinical improvement was observed within a few days. After further spinal MRI assessments, a reduction in longitudinal extension and thickness of medullary signal was observed, and hydrosyringomyelia was no longer present. Electromyography (EMG) showed signs of central neurogenic distress that was related to spinal cord impairment. The orthopedic evaluation ruled out osteoarticular anomalies, except for secondary scoliosis, involving the thoracolumbar curve with loss of cervical and lumbar lordosis and without any significant rotation of the vertebral bodies; physiotherapy was then prescribed. After discharge, the patient was monitored with clinical and instrumental follow-up, did not present with any neurological sequelae, and underwent custom rehabilitative treatment for six months, based on stabilization, mobilization, and specific postural exercises. Moreover, the patient was taught to perform auto-correction, spinal elongation, isometric exercise contraction, stabilizing exercises, and rotational breathing. At the 6-month follow-up, the patient presented good spine posture, no pain, restoration of physiological curves in the lateral view, absence of muscles contractures, and excellent balance.

## 3. Discussion

ATM is an immune-mediated central nervous system disorder classically described as demyelinating [8]. Mild spinal trauma or allergic status may be predisposing risk factors [7]. Clinical features of ATM may include pain, paresthesia, numbness, weakness (typically paraplegia or quadriplegia), and bowel and/or bladder dysfunction. The most common sequelae are sensory and bladder disturbances (15–50%). Approximately one-quarter are non-ambulatory or require walking aids, and 10–20% never regain mobility and/or bladder function [2]. The influence of age, time to onset of symptoms, and time to recovery varied in several studies. Studies in pediatric ATM have attempted to define risk factors for relapse and disability. A single-center study consisting of 47 children of younger ages (<3 years old) showed delayed time from symptom onset to treatment, higher spinal level involvement, radiologic evidence of longer segment, presence of T1-hypointense lesions, and lack of white cells in the CSF as predictors of disability [2]. A multicenter ATM study with retrospective ascertainment and longitudinal follow-up had a relapse frequency of 17%.

Clinical assessment is mandatory. In the proband, as reported in the literature, pain was the first symptom. On the other hand, the preservation of sensitivity is peculiar; in fact, the boy in our case did not present any alterations in sensitivity as confirmed by EMG or show any signs of central neurogenic distress. Intestinal and/or bladder dysfunctions were not present. During the acute phase, muscle tone and deep tendon reflexes in affected extremities may be decreased with typically increased tone and patellar areflexia over time. The lag time from symptom onset to maximal severity can considerably vary depending on the etiology and severity of disease and range from 4 h to 21 days [4]. Autonomic dysfunction is common, including variations in body temperature and instability in respiratory rate as well as heart rate and rhythm. Our patient presented flushing, hyperhidrosis, and itching, while cardiac and/or respiratory dysfunction were absent.

Imaging, especially MRI, is paramount for diagnosis. In fact, lesions are often located in the central cord and involve the gray and nearby surrounding white matter. Most of the lesions are longitudinal and extending and are characterized by involvement of three or more segments [9]. CSF findings consists of lymphocytosis (usually less than 100/mm^3^) and increased protein level (usually 100–120 mg/dL). In 20% to 50% of children with definite ATM, CSF analysis shows normal protein levels and white blood cells counts [1]. Evidence of intrathecal antibody synthesis, as demonstrated by positive oligoclonal bands and increased Immunoglobulin G (IgG) index, are suggestive of autoimmune myelitis. This pattern is observed most frequently with multiple sclerosis although other autoimmune in addition to infectious etiologies can also produce such findings [4].

Differential diagnosis includes several parameters: (1) other forms of myelopathy, such as compressive or non inflammatory disorders, including epidural hematoma, intervertebral disk herniation, vertebral body fracture, ischemic myelopathy due to arterial compromise or venous hypertension, and spinal cord tumors [10]; (2) secondary ATM, including infectious myelitis, rheumatological diseases (such as Systemic Lupus Erythematosus (SLE) and Sjogren’s syndrome [SS]), paraneoplastic syndromes, demyelinating CNS disease (ADEM, MS, and NMO) [11,12]; and (3) non-myelopathic disorders, such as Guillain-Barre syndrome.

Risk factors for disability are scale (A–C) of American Spinal Injury Association (ASIA) impairment at the onset of the disease, absence of CSF pleocytosis, spinal lesions as demonstrated with gadolinium enhancement, abnormal brain MRI findings, female sex, and absence of cervicothoracic lesions. 

Regarding the disability, the ASIA scale is internationally accepted for the measurement of spinal impairment, but it has rarely been applied to children. Other evaluation scales are the Expanded Disability Status Scale (EDSS), WeeFIM II system, clinician-derived motor recovery ordinal measures, and Paine and Byers scale (poor, fair, and good recovery) [2].

The International Standards for Neurological Classification of Spinal Cord Injury (ISNCSCI), which were initially developed for spinal cord injuries, is advisable for use in adults but also in children for detecting neurological impairment in people with spinal cord dysfunction [11].

ATM may have a relapsing course but is infrequent in pediatric ages; moreover, ATM could be categorized as relapsing ATM, a presentation of MS, part of a systemic autoimmune disease, or NMOSD in the setting of identified antibodies.

Standard first-line therapy in idiopathic ATM is a high dose course of iv corticosteroids that are prescribed as 30 mg/kg/die (maximum 1 g/day) of methylprednisolone for 3 to 7 days. IV methylprednisolone should be followed by an oral steroid starting at 1 mg/kg/day that is then tapered over 3 to 4 weeks [12]. 

In the proband, steroid therapy was effective and second-line therapies were not necessary. In the most severe cases of myelitis, it is advisable to use immunoglobulins and therapeutic plasma exchange for the form that is refractory to iv corticosteroids [12]. Most children have a good prognosis, with the exception of rapidly dramatic onset forms and those complicated by respiratory failure.

In the literature [12], rehabilitation has a great impact on disease evolution. There are a few reports on when and for how long physiotherapy should be done during childhood. In the proband, rehabilitative treatment was conducted for six months, based on stabilization, mobilization, and specific postural exercises. Moreover, the patient was taught to perform auto-correction, spinal elongation, isometric exercise contraction, stabilizing exercises, and rotational breathing. At the six months follow-up, in our study, the boy presented good spine posture, no pain, restoration of physiological curves in the lateral view, absence of muscles contractures, and excellent balance, which suggested that rehabilitative treatment made the healing process easier and helped prevent complications.

International collaborative studies for standardization of clinical assessment and investigation protocols are needed to better delineate prognostic factors for disability and relapse. There are no robust controlled trials in children to inform optimal treatment and ATM-related rehabilitation therapy.

## 4. Conclusions

The pediatric cases of ATM are less frequent than in adults but present a better prognosis. In the child in our case, onset of the symptoms was not preceded by an infectious episode but by a minor trauma. The young patient had good response to steroid therapy combined with physiotherapy. The rehabilitative treatment lasted for six months, and based on stabilization, mobilization, and specific postural exercises, allowed for complete healing. Collaboration among various specialists is worthwhile for making a correct and rapid diagnosis.

## Figures and Tables

**Table 1 jfmk-05-00039-t001:** Symptoms and findings in our patient.

Symptoms [1]	Patient
Pain	+
Paresthesias/Numbness	-
Weakness	+
Bowel/Bladder disfunction	-
**MRI** [7]	
Number of segments involved >3	+
Gray/White matter involvement	+
Enhancement	+
**CSF** [8]	
Increased IgG index	+
High liquor proteins	-
Pleocytosis	+

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
