# Peer review of "Secondary Scoliosis as a Complication of Acute Transverse Myelitis in a Child"

_jfmk, 2020, doi:10.3390/jfmk5020039_

Round 1

Reviewer 1 Report

This is a case report on a 11-years-child with diagnosis of acute transverse myelitis with associated pediatric scoliosis. The case represents secondary scoliosis as a complication of the myelitis. The manuscript is well written, with a good quality English.
I would underline some points

  • The introduction is essential in representing the aspects of pathology. I suggest to provide more information about differential diagnosis at line 51. It is possible to take the paragraph in discussion (127-133).
  • The discussion reaches the information about pathology and treatment.  

At Line 113, Patellar reflexia (do you mean “areflexia” or “reflexes”?)
Are there other cases of secondary scoliosis with ATM in literature?
Which was the importance of neurosurgeon in this pathology?

Author Response

Dear Reviewer,

Thank you for your comments and suggestion.

Q1) The introduction is essential in representing the aspects of pathology. I suggest to provide more information about differential diagnosis at line 51. It is possible to take the paragraph in discussion (127-133)

A1) Following your evaluable suggestion, the paragraph was modified. References were updates.

Q2) The discussion reaches the information about pathology and treatment.  At Line 113, Patellar reflexia (do you mean “areflexia” or “reflexes”?)

A2) Sorry for the mistake, we mean areflexia. The word was corrected.

Q3) Are there other cases of secondary scoliosis with ATM in literature?

A3) There are no similar cases in literature. To our knowledge, this is the first cause of myelitis associated to secondary scoliosis.

Q4) Which was the importance of neurosurgeon in this pathology?

A4) Neurosurgery actually denied his role in treatment of this pathology

Reviewer 2 Report

That you for submitting this unusual case report. Was there any similar case reported?

This boy already had a scoliosis, no lordosis, and a mild C3 anterolysthesis. This, in addition to the hydrosyringomyelia, suggests a dysraphism syndrome. Was there an Arnold-Chiari anomaly, and/or a spina bifida occulta, or other malformations?

I do not think that the fall caused the medullary lesion. I think he fell because of his spinal lesion. The abruptness of the onset suggests a vascular etiology, maybe a partiel occlusion of the anterior spinal artery, in view of the absence of sensory deficits (although there was clear allodynia), possibly an impairment of the posterior vasculature. Did you rule out a dural fistula?

A few questions: what are hyposthenia and rotational breathing?

You mention a high CSF IgG Index; did you test for oligoclonal bands?

I suggest you abridge both the introduction and the discussion.

Author Response

Dear Reviewer,

Thank you for your comments and suggestions about the submitted manuscript.

Q1) That you for submitting this unusual case report. Was there any similar case reported?

A1) There are no similar cases in literature. To our knowledge, this is the first cause of myelitis associated to secondary scoliosis.

Q2) This boy already had a scoliosis, no lordosis, and a mild C3 anterolysthesis. This, in addition to the hydrosyringomyelia, suggests a dysraphism syndrome. Was there an Arnold-Chiari anomaly, and/or a spina bifida occulta, or other malformations?

A2) No other malformations were associated. MRI did not show any alterations. No syringomelia or Arnold-Chairi anomaly have been noted.

Q3) I do not think that the fall caused the medullary lesion. I think he fell because of his spinal lesion. The abruptness of the onset suggests a vascular etiology, maybe a partiel occlusion of the anterior spinal artery, in view of the absence of sensory deficits (although there was clear allodynia), possibly an impairment of the posterior vasculature. Did you rule out a dural fistula?

A3) Dural fistula was ruled out at MRI examination. It is possible your theory about a partial occlusion of the anterior spinal artery, but it is not clearly demonstrable. In fact, MRI did not show any vascular anomalies.

Q4) A few questions: what are hyposthenia and rotational breathing?

A4) With hyposthenia we mean weakness in walking, diminished strength and tonicity. With rotational breathing we mean a group of spinal extension, lateral spinal flexion and rotational exercises performed in synchrony with deep breathing.

Q5) You mention a high CSF IgG Index; did you test for oligoclonal bands?

A5) We tested oligoclonal bands with negative results. This was inserted in the text

Q6) I suggest you abridge both the introduction and the discussion.

A6) Following your indication, a part of discussion was moved to Introduction.

Round 2

Reviewer 2 Report

Thank you for your reply.

Plain MRI is insufficient to rule out vascular etiology for spinal cord disorders.

I still think that a vascular etiology is a strong possibility, given the abruptness of the onset.